

# Optimal control simulations tracking wearable sensor signals provide comparable running gait kinematics to marker-based motion capture

Grace McConnochie[1], Aaron S. Fox[2], Clint Bellenger[3] and Dominic Thewlis[1]

[1] Centre for Orthopaedic & Trauma Research, Adelaide Medical School, University of Adelaide, Adelaide, South Australia, Australia
[2] Institute for Physical Activity and Nutrition, School of Exercise and Nutrition Sciences, Deakin University, Geelong, VIC, Australia
[3] Alliance for Research in Exercise, Nutrition and Activity (ARENA), Allied Health and Human Performance Unit, University of South Australia, Adelaide, South Australia, Australia

Corresponding author
Grace McConnochie,
grace.mcconnochie@adelaide.edu.au

## ABSTRACT

**Objective:** Inertial measurement units (IMUs) offer a method for assessing gait beyond the confines of a laboratory. Signal noise and calibration errors pose significant obstacles to accurately estimating joint angles, particularly during dynamic activities such as running. Advancements in dynamic optimisation tools could enable a more comprehensive analysis with fewer sensors and/or low-quality data. The objective of this study was to compare two IMU-based modelling approaches (inverse kinematics and optimal control simulations) with optical marker-based motion capture in reconstructing running gait kinematics.

**Methods:** Six participants performed treadmill running at three speeds whilst marker trajectories and IMU signals were collected concurrently. The subject-specific biomechanical model consisted of a 3D representation of the lower body and torso, with contact spheres added to simulate ground contact in the optimal control simulations. The objective of the optimal control simulations was to track the accelerations, angular velocities, and orientations of eight sensors with simulated signals from the model sensors. Additional constraints were enforced, reflecting physiological and biomechanical principles and targeting dynamic consistency. The objective of the IMU-based inverse kinematics was to minimize the difference between the input and simulated sensor orientations. The joint kinematics derived from both methods were compared against optical marker-based motion capture across a range of running speeds, evaluating the absolute and normalized root mean square errors.

**Results:** Compared with motion-capture joint angles, optimal control simulations resulted in lower absolute errors (RMSE 8° ± 1) that were consistent across all speeds. IMU-based inverse kinematics exhibited greater differences with motion capture (RMSE 12° ± 1), which was more significant at faster speeds. The largest absolute inaccuracies were observed in the sagittal angles when not normalizing for the joint range of motion. The computational times for the optimal control were 46 ± 60 min, whereas they were 19.3 ± 3.7 s for the IMU-based inverse kinematics.

**Conclusions:** Compared with traditional IMU-based inverse kinematics, the optimal control approach provides a more comparative representation of joint kinematics from optical motion capture. This method can mitigate errors associated with closely tracking IMU noise and drift, and it offers a dynamic analysis that considers the underlying forces and torques producing movement. However, these advantages come at the expense of challenges in parameter selection and computational cost.
**Significance:** These findings highlight the potential of using IMUs with optimal control methods to provide a comprehensive understanding of gait dynamics across diverse applications. IMU-based inverse kinematics remains a viable option for faster computation and when model fidelity is less of a concern.

## INTRODUCTION

Inertial measurement units (IMUs) are becoming increasingly common for assessing human movement, including running gait (*Blazey, Michie & Napier, 2021*; *Benson et al., 2022*; *Zeng et al., 2022*; *García-de-Villa et al., 2023*). These small, lightweight, and inexpensive devices consist of an accelerometer, gyroscope, and in some cases, a magnetometer. They can be used outside of laboratory constraints, providing a more ecologically valid analysis. Despite their numerous benefits, IMU-based methods for evaluating joint kinematics also present challenges. Sensor drift and noise, integration errors, and device calibration issues pose significant challenges when deriving joint kinematics from sensor signals, particularly in dynamic movements.

Sensor fusion methods (*Picerno, 2017*; *Weygers et al., 2020b*) or machine learning techniques (*Gholami et al., 2020*; *Rapp et al., 2021*; *Xiang et al., 2022*) can be used to derive body segment and joint kinematics. To minimize error, researchers have applied sophisticated algorithms to filter and fuse sensor data and/or, constrain the outputs *via* computational biomechanical models. These models can be combined with inverse kinematic methods, minimizing the errors between the orientations of experimental IMUs and analogous IMU frames on the model, while being subject to physiologically feasible joint constraints. While kinematic outcomes have been validated with this approach, *Ferrari et al. (2010)*, *Zhang et al. (2013)*, *Tagliapietra et al. (2018)*, *Karatsidis et al. (2019)*, *Cereatti et al. (2024)* limiting an analysis to joint kinematics overlooks the interplay between energetics, and internal and external loads in relation to gait. Biomechanical models offer the ability to estimate other outcomes, including muscle or actuator activation dynamics, joint and ground reaction forces, and locomotive cost (*Delp et al., 2007*; *Karatsidis et al., 2019*; *Slade et al., 2021*; *Lloyd, 2021*). However, many applications of musculoskeletal modelling are based on optical motion capture methods in a laboratory setting (*Dorn, Schache & Pandy, 2012*; *Apte, 2021*).

Parameters, including kinematic measures and external and/or internal loads, are commonly computed in a consecutive manner using inverse kinematics, inverse dynamics,

and static optimization or computed muscle control approaches (*Delp et al., 2007*; *Karatsidis et al., 2019*). Inverse methods depend critically on input data, with the potential for error accumulation across steps. The temporal independence of inverse kinematics may result in unrealistic joint angle fluctuations, as solving for each time step separately can lead to closely tracking noise or abrupt changes in the input data. Employing inverse dynamics to estimate joint moments and internal forces introduces dynamic inconsistencies, necessitating residual forces and moments in the solution (*Faber, van Soest & Kistemaker, 2018*). Additionally, inverse dynamics relies on directly measured external loads, limiting data acquisition to environments where the ground reaction force is measurable. Finally, static optimization proves suboptimal for modelling muscle forces during highly dynamic activities, failing to account for the tendon compliance that is pertinent in fast running (*Lin et al., 2012*).

One approach to reducing the dependence of the modelled kinematics on the accuracy of the experimental data and its associated error propagation is to use biomechanical models within an optimal control framework (*Dembia et al., 2019*). Optimal control methods solve for the control variables that minimize or maximize an objective function. In biomechanics, the objective can include minimizing errors between experimental and modelled data and/or locomotor objectives such as minimizing muscular effort, the cost of transport or mechanical loading (*van den Bogert, Blana & Heinrich, 2011*; *Wang et al., 2012*; *Lin, Walter & Pandy, 2018*). A more dynamically consistent simulation can be obtained in a single trajectory optimization, minimizing error propagation and the use of residual forces or moments (*van den Bogert, Blana & Heinrich, 2011*; *Fluit et al., 2014*; *Lee & Umberger, 2016*; *De Groote et al., 2016*). This method is highly suitable for solving problems with unstable dynamics as seen in running gait (*Dorschky et al., 2019b*; *Nitschke et al., 2020*; *Haralabidis et al., 2021*; *Hosoi & Fay, 2024*), while offering the ability to model other metrics not available with inverse kinematic methods (*e.g.*, internal forces and torques, and energy cost, with ground contact models an option in lieu of externally measured loads).

The use of optimal control in biomechanics has burgeoned with advancements in computational power, available toolboxes, and methodological approaches (*De Groote & Falisse, 2021*; *Febrer-Nafría et al., 2023*; *Hosoi & Fay, 2024*). Tracking IMU signals with musculoskeletal models, combined with physiologically relevant cost functions, represents a potentially novel method to assess locomotion in a more holistic, dynamically consistent, and individualised manner (*Dorschky et al., 2019a*). Recent work has tracked acceleration and angular velocity signals with 2D musculoskeletal models in optimal control simulations. Minimizing the error between experimental IMUs and simulated signals from model IMUs was able to accurately simulate walking and running gait (*Dorschky et al., 2019a*). Whilst pioneering work, the use of a two-dimensional model limited the analysis to the sagittal plane, neglecting changes in some joint angles commonly assessed in running gait analysis, such as pelvis or hip rotations (*Ceyssens et al., 2019*; *Vannatta, Heinert & Kernozek, 2020*).

The aim of this study was to compare biomechanical simulations using two IMU-based methods with those from optical marker-based motion capture. Three-dimensional joint

angles derived from optimal control simulations, which used raw IMU signals as tracking inputs, were compared with those obtained from optical marker-based motion capture. Additionally, the joint angles computed with IMU-based inverse kinematics were compared with motion capture angles. It was hypothesised that the optimal control method would better model joint kinematics compared with an inverse IMU-based method that is more contingent on IMU data accuracy, *Borno et al. (2022)* and that the level of agreement would be within the range of marker registration and scaling error associated with motion capture (*Uchida & Seth, 2022*). It was anticipated that employing an optimal control approach, while offering greater fidelity in the outcomes, would incur greater computational costs and time.

## METHODS

The University of Adelaide's Human Research Ethics Committee approved the protocol for this observational study (#H-2022-120). All participants were fully informed of the experimental procedures and any associated risks and provided written informed consent prior to enrolment in the study. A graphical overview of the study method is shown in Fig. 1.

### Participants

Six adult experienced distance runners (three male, three female) were recruited for the study from local running clubs and social media pages. The small sample size was due to the significant computational time involved for each participant, and is comparable to that of optimal control-based studies, with sample sizes ranging from 1–10 (*Nitschke et al., 2020*; *Haralabidis et al., 2021*; *Veerkamp et al., 2021*; *Falisse, Afschrift & Groote, 2021*; *Nitschke et al., 2023*). The inclusion criteria were as follows: (i) Individuals aged between 18 and 50 years, (ii) BMI between 18–25. (iii) Completion of a minimum average of 50 km/week of running over the past 3 months, as self-reported or obtained from logged training runs *via* a GPS watch.

Participants were excluded if they met any of the following criteria: (i) Presence of neurological, cardiovascular, or musculoskeletal conditions; (ii) Inability to provide informed consent; (iii) Incidence of any running-related injury over the past month that necessitated the participant to miss three or more consecutive training runs.

The participants were instructed to continue with their usual physical training throughout the testing period, but to abstain from intense exercise 24 h prior to the experimental sessions and avoid unaccustomed activities that may cause abnormal levels of physical stress.

### Experimental setup
#### *Motion capture*

A lower body and torso marker set consisting of 32 reflective markers on bony landmarks of the lower limb, pelvis and trunk (*Cappozzo et al., 1995*) was fitted to each participant. To minimize movement or detachment, markers were attached using superglue on the skin locations and double-sided adhesive tape secured with hyperfix tape over clothing or
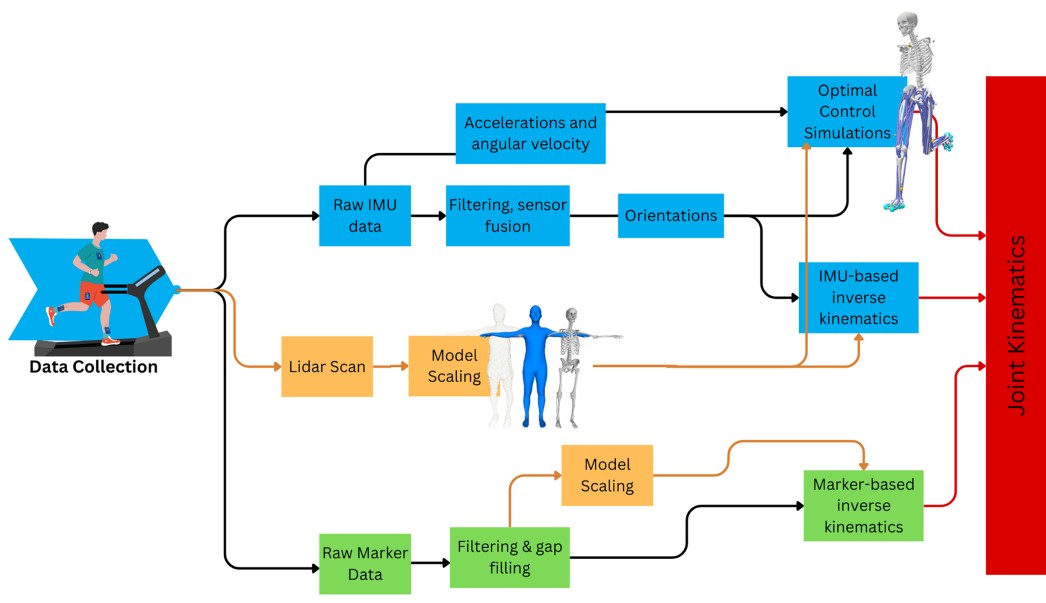

**Figure 1** Overview of study workflow to obtain joint kinematics from two IMU-based methods and compare outcomes with optical marker-based motion capture.

footwear. Marker trajectories were collected *via* a 10-camera Vicon Vantage V5 motion capture system (Vicon 4.2 Motion Systems, Oxford Metrics, Oxford, UK) at 100 Hz.

### IMU sensors

Eight IMeasureU Blue Trident sensors (Vicon Motion Systems Ltd, Oxford, UK) were placed on the participants at recommended locations (*Apte, 2021*; *Scalera et al., 2021*). Sensors were placed on the sacrum and sternum, and the anterior-medial tibia, lateral mid-thigh, and proximal aspect of the shoe of both limbs (Fig. 2). Sensors were secured using hyper-fix and strapping tape to limit soft-tissue artefacts (*Johnson et al., 2020*). A separate custom-made housing for the pelvis sensor secured it to the participant's waistband. Acceleration, magnetometer and angular velocity signals were collected at 1,125 Hz.

## Data collection

Prior to commencing the running trial, two static trials were collected to establish a sensor coordinate system in the global frame. First, a static trial was performed in a neutral, standing pose, followed by sitting on the edge of a chair while leaning back with outstretched legs. A Light Detection and Ranging (LiDAR) point cloud of the participant was obtained using an Apple iPad with the Scaniverse application ("*Toolbox AI, 2023*") to inform the model scaling process (*McConnochie et al., 2025*). All running sessions were conducted on a motorized treadmill (Woodway Pro) located in a controlled laboratory environment. The participants wore their own attire and footwear typical for a race or high-effort running session. After completing a 10-min warm-up at a self-selected speed, data were collected continuously for each participant running for three minutes at three speeds. The collection of marker and IMU data was synchronised. Speeds were set based

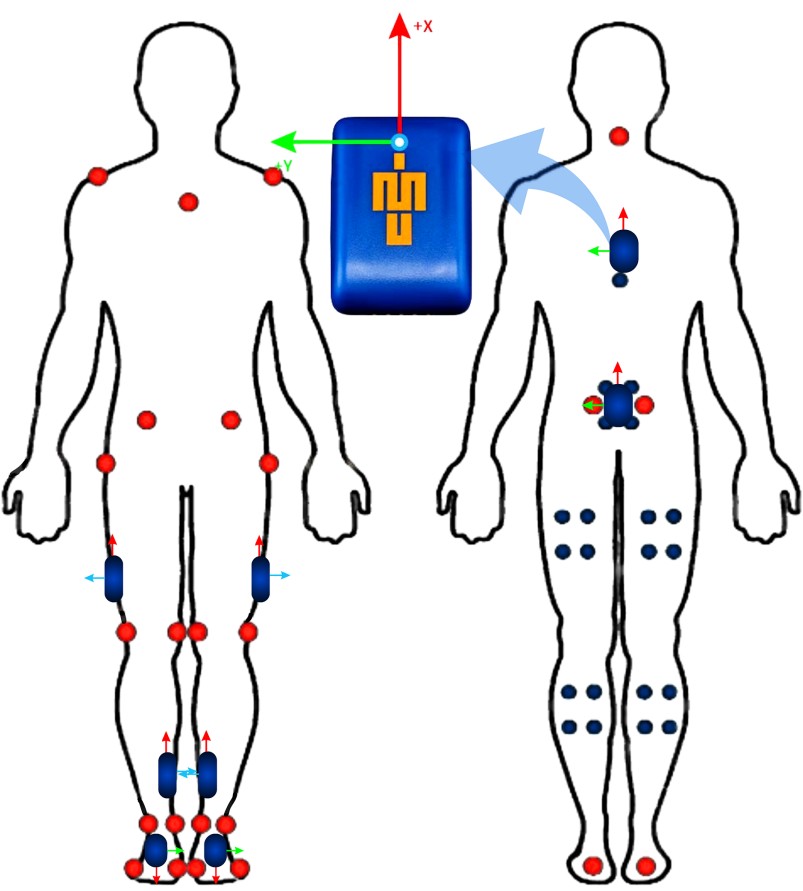

**Figure 2 IMU and marker placement.**               

on individual ability, at a comfortable pace that minimized fatigue. Those achieving a 5 km personal best under 20 min ran at 10, 12, and 14 km/hr, with others running at 8, 10, and 12 km/h, to represent a comparative range of steady training pace (*Daniels, 2013*).

## Signal processing
### *Motion capture*
The three-dimensional coordinates of the markers were reconstructed using Vicon Nexus (v2.16). Spurious or noisy data points not corresponding to the participant's body were removed. Marker processing included labelling, tracking, and gap-filling, with a rigid-body fill applied where possible. The trajectories were filtered with a low-pass Butterworth filter with an 8 Hz cut-off. A custom MATLAB (The MathWorks, 2022b) script generated the OpenSim input files.

### *IMU sensors*
The acceleration and angular velocity signals to be tracked in the optimal control simulations were filtered with a low-pass, 4th-order Butterworth filter with a cut-off frequency of 10 Hz to reduce the effects of high-frequency noise (*Weygers et al., 2020a*). Gyroscope drift was removed by linear fitting the data and detrending any measured offset of magnitude greater than 0.001 rad/s (*Bailey et al., 2021*).

A sensor fusion algorithm was used to estimate sensor quaternions in the global space, using raw acceleration, angular velocity and magnetometer signals (*Laidig & Seel, 2022*). Due to the heading offset and potential magnetic interference observed in pilot testing, the raw IMU signals were first rotated in 90° increments to approximately align all the sensor axes with both the global and body segment coordinate systems based on their approximate orientation relative to the participant in the static pose. The mean heading angle of the sensor over each trial was set to 0°. For the foot and lower leg IMUs, orientations outside the sagittal plane were set to zero to account for the two degrees of freedom ankle and knee joints.

The body segment-to-sensor transformation was established using a functional alignment calibration method (*Palermo et al., 2014*) with IMU data from both static poses. Two vectors in the sagittal plane were determined from the sensor gravity vector during each pose, calculated as the mean of the rotated accelerometer data over a stationary, noise-free period. The superior-inferior (SI) axis of each sensor was defined solely by the standing gravity vector. The medial-lateral axis (ML) results from the cross-product of the two gravity vectors from each pose. The anterior-posterior (AP) axis was solved from the SI and ML vector cross-product. A final cross-product between the AP and SI axes redefined the ML axis to ensure orthogonality. The sensor axes coordinate vectors were converted to quaternions, with a heading set to 0°, establishing the orientation of the sensors on the model in the static pose.

Stride segmentation was determined from the angular velocity of the lower leg IMU about the sagittal plane (*Ben Mansour, Rezzoug & Gorce, 2015*). Five strides were extracted at the beginning, middle and end of each speed interval, resulting in 15 strides analysed at each speed, and 45 for each participant (*Souza Oliveira & Pirscoveanu, 2021*). The IMU signals were segmented based on these time points for input into each optimal control simulation.

## Data analysis
### Model calibration and scaling
The simulations used a previously developed 3D OpenSim model (*Rajagopal et al., 2016*) without upper limbs, and modifications to allow for fast-paced running (*Lai, Arnold & Wakeling, 2017*). The model contained 23 degrees of freedom (DOF), comprising of six DOF ground to-pelvis joint, three DOF for the lumbar and hip joints, and one DOF knee, ankle, metatarsalphalangeal and subtalar joints.

The OpenSim models were scaled in accordance with each participant's anthropometry in MAP Client (*Zhang et al., 2014*) generating body segment scale factors *via* an atlas-based statistical shape modelling process informed by skeletal anatomical landmarks (*Bakke & Besier, 2020*; *Akhundov et al., 2022*). For the marker-based inverse kinematics, anatomical landmarks from key marker positions during the static trial were used to generate the subject-specific scale factors for the OpenSim model body segments. For the model using IMUs, an independent scaling method that could be implemented without marker trajectories was developed. A Skinned Multi-Person Linear model (SMPL) model

was fitted to the LiDAR scan point cloud using a Basis Point Set method, *Prokudin, Lassner & Romero (2019)* from which an anatomical skeleton and its landmarks were inferred (*Keller et al., 2022*; *McConnochie et al., 2025*).

## Inverse methods

The inverse kinematics problem can be formulated as:

$$\text{Minimize}: \quad J(\theta) = \sum_{i=1}^{N} \sum_{t=1}^{T} w_i \cdot ||\mathbf{x}_{\text{exp},i}(t) - \mathbf{f}(\mathbf{u}(t), \theta)||^2 \tag{1}$$

where $N$ is the number of inputs, in this case, marker trajectories or IMU sensor orientations. $T$ is the number of frames or time steps, $J(\theta)$ is the cost or objective function, $w_i$ is a weighting factor for the given input, $\mathbf{x}_{\text{exp},i}(t)$ is the experimental position of the input at time $t$. Lastly, $\mathbf{f}(\mathbf{u}(t), \theta)$ is the model-predicted position of the input at time $t$ based on the generalized coordinates $u(t)$ and joint angles $\theta$. As the data were of insufficient resolution to accurately capture subtalar and metatarsophalangeal joint (MTP) kinematics in the inverse simulations, these joints were fixed.

Simulations were solved for each participant and speed using marker and IMU-based inverse kinematics over the entire running trial. Strides were then segmented using the time points determined from the lower-leg IMU angular velocity. Three degree of freedom pelvis and hip angles, and knee flexion, and ankle flexion angles were extracted over each of the 15 strides at each speed for comparison with the IMU-based methods. All kinematic data were time-normalized to 0% to 100% of the gait cycle, with reference to the right footstrike.

### Motion-capture inverse kinematics

Using the marker-scaled model, optical marker-based motion capture joint kinematics were computed over the entire trial using the inverse kinematics solver in Opensim v4.4 (*Seth et al., 2018*). Joint angles were calculated by minimizing the sum of the squared differences between the positions of virtual markers on the model and experimental marker trajectories. The weighting factors for each marker were determined heuristically. Those markers with a greater tendency towards soft tissue artefacts and/or placed on clothing (*i.e.*, pelvis and thigh markers) had a lower contribution to the objective. An exemplary inverse kinematics settings file is provided at https://doi.org/10.5281/zenodo.14796986.

### IMU inverse kinematics

IMUs were placed on the LiDAR-scaled OpenSim model with the pitch and yaw determined from the static calibration poses, and a heading of 0°. Differences between the experimental sensor orientations and the orientations of the virtual IMU frames placed on this model were minimized using the OpenSense (*Borno et al., 2022*) inverse kinematics solver in OpenSim v4.1. Equal weighting factors were applied to each sensor in the objective function.

## Optimal control simulations

IMU data tracking simulations of running gait were formulated as optimal control problems which were solved using direct collocation in OpenSim Moco (version 4.4.1) (*Dembia et al., 2019*) in Python 3.8 on a Phoenix High-Performance Computer (256 GB RAM, 72 CPU).

The models were identical to those used in the IMU-based inverse method, except that in the optimal control simulations, the model joints were driven by torque actuators with an activation time constant of 0.01 s. The optimal torques ranged from 50 to 500 Nm depending on the joint and degree of freedom, with a control activation limit of 5. Additional residual actuators were added to the pelvis rotational and vertical translation degrees of freedom with optimal values of 1 NM and 25 N respectively.

Contact between the foot and the ground was modelled using 11 ground-contact spheres on each foot, approximating a previously published Hunt-Crossley foot-ground contact model (*Hunt & Crossley, 1975*; *Serrancolí et al., 2019*). The spheres were placed evenly across the plantar surface in proportion to the foot size (Fig. 3). The static and dynamic friction and transition velocity properties were consistent with reference values of 0.8 and 0.5 respectively. The modulus (3.06 MPa) and damping coefficient (2.0 s/m) of the contact elements modelled the deformation and energy return of the heel region of a human foot in an athletic shoe (*Aerts & De Clercq, 1993*). Passive damping was added to the lower limb and lumbar joints, to model ligaments and other passive structures (*Anderson & Pandy, 2001*). To improve the modelled foot-ground contact mechanics (*Falisse, Afschrift & Groote, 2021*) the MTP joint was unlocked in these simulations, with a linear rotational spring force with a stiffness of 25 Nm/rad added to represent the MTP joint passive structures (*Sasaki, Neptune & Kautz, 2009*).

### Initial guess

Optimization problems require an initial guess that serves as a starting point for the solver to search the solution space. A suitable initial guess is pertinent for optimal control modelling to avoid excessive computational effort or convergence on a local minimum that does not resemble the desired motion. As such, a guess was generated from a tracking simulation of generic kinematics from running at 14.4 km/h (*Nitschke et al., 2020*).

### Objective equation

The optimal control problem can be formulated as follows: Minimize $J(x, u)$ subject to the following constraints:

$$\dot{x} = f(x(t), u(t))$$
$$x(0) = x_0 \tag{2}$$
$$u(t) \in U, \quad \forall t \in [0, T].$$

In this equation, the objective function $J(x, u) = J_{effort} + J_{trackIMU} + J_{trackIK} + J_{GRF}$ represents the cost to be minimized over the time period $T$. The primary objective of these simulations was to determine the model states $x$ and controls $u$ that minimize errors in tracking experimental data, subject to the system dynamics and constraints. The objective function consisted of the following terms each with individual weightings $w_{Ni}$
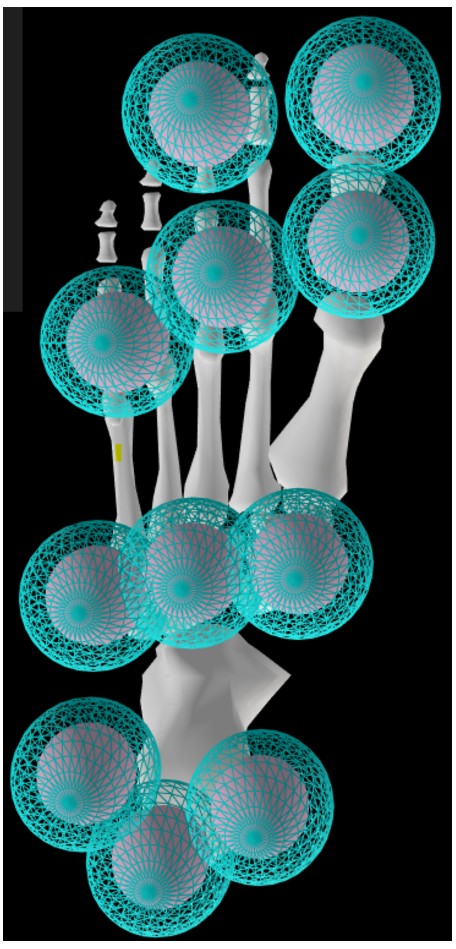

**Figure 3 OpenSim model foot contact geometry.**

As the primary objective, the 3D orientations, accelerations and angular velocities of the input IMUs are tracked with the signals from the analogous virtual IMUs placed on the model (Eq. (3)).

$$J_{\text{track IMU}} = \int_0^T \left[ \frac{1}{N_1} w_{N1} \sum_{k=1}^{N_1} \sum_{i=x,y,z} \left( \frac{a_{i,k}(t) - \mu_{a_{i,k}}(t)}{\sigma_{a_k}} \right)^2 + \left( \frac{\omega_{i,k}(t) - \mu_{\omega_{i,k}}(t)}{\sigma_{\omega_k}} \right)^2 + \left( \frac{\theta_{i,k}(t) - \mu_{\theta_{i,k}}(t)}{\sigma_{\theta_k}} \right)^2 \right] dt \quad (3)$$

where $N_1$ is the number of IMUs (8) and $a$, $w$, and $\theta$ represent the accelerations and angular velocities of the IMU frame and orientations in the global frame in the $x$, $y$ and $z$ coordinates respectively. $\mu$ is the corresponding signal from the virtual sensor on the model. Differences were normalized to the measurement standard deviation $\sigma$.

A secondary aim was to minimize effort (*Miller et al., 2012*; *Miller & Hamill, 2015*; *De Groote & Falisse, 2021*; *Veerkamp et al., 2021*) (Eq. (4)).

$$J_{effort} = \frac{1}{D} \int_0^T \left[ w_{N3} \sum_{i=1}^{N_3} \mu_i^3(t) \right] dt \quad (4)$$

where D is the distance travelled by the model's centre of mass, $\mu_i(t)$ is the value of the control actuator at time $t$ and $N_3$ is the number of actuators. A larger weighting $w_3$ was applied to the model's residual actuators in the effort goal term to penalize their use.

An objective tracking generalized joint kinematics from a dataset of fast running kinematics was added to act as a regularisation term, to assist with convergence and noise reduction (*Nitschke et al., 2020*) (Eq. (5)).

$$J_{trackIK} = \frac{1}{T} \int_0^T \left[ \frac{1}{N_2} w_{N2} \sum_{i=1}^{N_2} (x_i(t) - \mu_i(t))^2 \right] dt. \tag{5}$$

Here, $N_2$ is the number of joint kinematics and $\mu_i(t)$ is the value of the input state variable at time $t$. An additional tracking goal was added to track the knee and hip flexion angles from the IMU inverse kinematics solved for in OpenSense, as this was found to improve joint angle agreement in the sagittal plane in pilot testing.

A generalized ground reaction force trajectory in the horizontal and vertical directions from the same dataset was tracked (*Nitschke et al., 2020*) to improve the model foot contact with the ground. The error between the force trajectory scaled in proportion to the participant's bodyweight and ground reaction forces modelled from the contact sphere, were minimized in the vertical and horizontal directions (Eq. (6)).

$$J_{GRF} = \frac{1}{W} \int_0^T \left[ w_{N4} \sum_{j=x,z} \left( GRF_j(t) - GRF_{j,\text{ref}}(t) \right) \right] dt \tag{6}$$

where $GRF$ is the ground reaction force from the modelled contact spheres at time $t$, and $GRF_{ref}$ is the reference force. $W$ is the model's body weight.

The optimal control formulation weights $w$ for each of these objectives were determined heuristically. The values were chosen to minimize the sum of Pearson correlations when comparing model kinematics with the analogous experimentally tracked/measured data on a subset of five strides from each participant. The weighting factors for the acceleration, angular velocity and orientations of the IMUs were 0.002, 0.01 and 3, respectively. The joint kinematics tracking, effort minimization, and ground reaction force goal weighting factors were 1, 0.5 and 1,000, respectively. Note that due to the different magnitudes and units of the goal inputs, these weighting factors do not represent the relative contribution to the overall simulation objective. Instead, these weightings were chosen with the aim of having approximately 90% of the objective comprised of the tracking of each of the three IMU signals, with the remaining contribution from the auxiliary goals. An example optimal control setup file is available at https://doi.org/10.5281/zenodo.14796986.

Bounds on the time taken to complete a full gait cycle, along with constraints on problem states and controls, were applied to limit the solution space and ensure that the solution was within a feasible range for human running locomotion. The state bounds were set to to the joint ranges of the optical motion capture reference data $\pm 5$–$10°$, depending on the variable. A CasADi solver (*Andersson et al., 2019*) within OpenSim Moco was used

to solve the predictive simulations. Convergence and constraint tolerances were set to 0.001 and 0.005 respectively.

## Statistical analysis

Joint kinematics were extracted in the body and joint reference systems, based on ISB conventions (*Wu et al., 2002*). The root mean square error (RMSE) was computed for each participant over each time normalized gait cycle, comparing data from marker-based kinematics with each of the optimal control simulations and IMU-based kinematics. To allow for relative comparisons between different motion planes, the RMSE was normalized by the average joint range of motion for each participant at each speed (*Ren, Jones & Howard, 2008*). All the data are reported as the mean ± SD.

To ensure the integrity of the dataset, optimal control simulations with a RMSE exceeding 30° for any given joint angle were subjected to visual inspection. If a simulation displayed anomalous joint angle trajectories, the simulation was inspected qualitatively, and if it was deemed to have converged on an unrealistic solution (*i.e.*, did not resemble a running gait pattern), it was excluded from further analysis. Any simulation that failed, or did not converge on a solution within 15 h or 7,000 iterations, was also excluded from the analysis.

## RESULTS

The participant characteristics and their personal best times are outlined in Table 1

Out of the 270 total simulations, 266 successfully converged on a solution. Among them, three were identified as outliers, involving Participant 3 running at medium and fast speeds. The mean simulation time across all successful optimal control simulations was 46 ± 60 min. The longest time was 8 h and 46 min, again for Participant 3 (Fig. 4A). IMU-based simulation and marker-based inverse kinematics had much shorter and more consistent computational times, with average solve times of 19.3 ± 3.7 and 3.7 ± 0.4 seconds per stride respectively (Fig. 4B).

Compared with marker-based inverse kinematics, the mean (± SD) RMSE across all joint angles was 7° ± 1° in the optimal control simulations, ranging from 4°–12°. The errors ranged from 4°–26°, with a mean of 10° ± 1° using IMU-based inverse kinematics (Table 2A, Fig. 5). Errors in the optimal control simulations did not differ in magnitude with faster speeds (Fig. 6A). This finding was also seen in the IMU-based inverse kinematics (Fig. 6B), excluding knee flexion where errors were greater at slower speeds (Fig. 5). When normalising differences to the joint range of motion, lumbar extension still exhibited the greatest differences comparing the optimal control approach with motion capture. The normalized errors for the IMU-based kinematics were greatest for pelvic tilt (Table 2B).

Qualitative comparisons between optimal control simulations and marker-based inverse kinematics revealed regions of notable difference for lumbar extension over the gait cycle, and periods of hip and knee flexion (40–80% gait cycle), and ankle flexion (0–20% gait cycle) (Fig. 7). When comparing OpenSense IMU inverse kinematics with

**Table 1 Participant characteristics, including best times over the 5 km (a) or 10 km (b) distance, and experimental treadmill running speeds.**

| Participant | Gender | Body mass (kg) | Height (cm) | BMI (kg. $m^2$) | $5^a$/$10^b$ km best time | Running speeds (km/h) |
|---|---|---|---|---|---|---|
| 1 | M | 59.2 | 173 | 19.8 | $17{:}26^a$ | 10,12,14 |
| 2 | F | 72 | 172.5 | 24.2 | $37{:}35^b$ | 10,12,14 |
| 3 | M | 82.1 | 186 | 23.7 | $19{:}15^a$ | 10,12,14 |
| 4 | F | 51.2 | 162 | 19.5 | $21{:}54^a$ | 8,10,12 |
| 5 | F | 61.4 | 164 | 22.8 | $21{:}06^a$ | 8,10,12 |
| 6 | M | 80.2 | 178.5 | 25.2 | $18{:}36^a$ | 10,12,14 |

marker-based inverse kinematics, these errors tended to be larger in magnitude, with regions of greater error in the lumbar and pelvis joints seen outside the sagittal plane.

## DISCUSSION

This study compared two IMU-based modelling approaches for reconstructing joint motion: optimal control simulations seeking to minimize differences between experimental and simulated IMU signals, and IMU-based inverse kinematics. A 3D, subject-specific biomechanical model was used in both methods to model running gait across various speeds. Optical marker-based motion capture was employed as the comparative gold standard to assess the accuracy of the modelled joint kinematics.

This study confirmed the hypothesis that an optimal control modelling approach more closely resembles the joint kinematics obtained from optical motion capture than traditional inverse-based methods using IMUs. The average RMSE was found to be 7° and 10° for each of the methods, with a consistent level of agreement observed across all speeds, where the mean errors within participants did not differ by more than 2°. The lowest errors with marker-based inverse kinematics in both methods were in joint angles outside the sagittal plane (Fig. 5) IMU-based inverse kinematics had greater errors than optimal control simulations in 10 out of 11 joint angles, with magnitudes up to 26° seen in knee flexion (Table 2). Conversely, when the joint range of motion was considered, normalised errors were smallest in knee flexion for the optimal control simulations compared with motion capture (Table 2B). These results underscore the ability of an optimal control approach to effectively capture three-dimensional motion at moderate to fast running speeds.

While optimal control generally showed better agreement with motion capture than did IMU-based inverse kinematics, this was not observed for the lumbar flexion degree of freedom. A fixed offset between joint angle outcomes, with the optimal control simulations having a more upright torso, was not observed between marker and IMU-based inverse-kinematic approaches. Optimal control simulations must also satisfy system dynamics to arrive at a solution. With the torso being a large and heavy model body segment, reducing lumbar flexion would minimize the need for high actuator forces to maintain any forward lean. The necessity of maintaining an upright posture may therefore come at the expense of tracking lumbar flexion. In the IMU-based kinematics the greatest difference was observed in the pelvic tilt degree of freedom. This fixed offset observed in

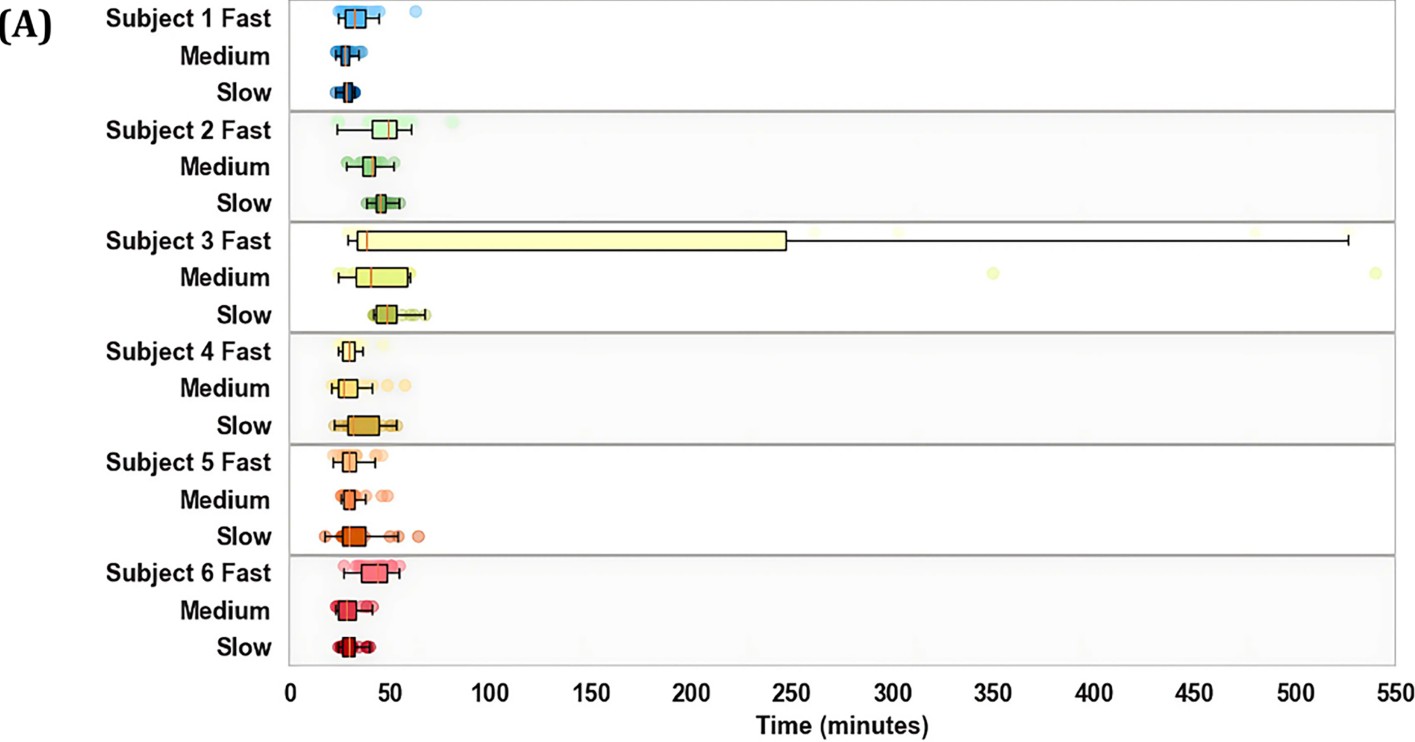

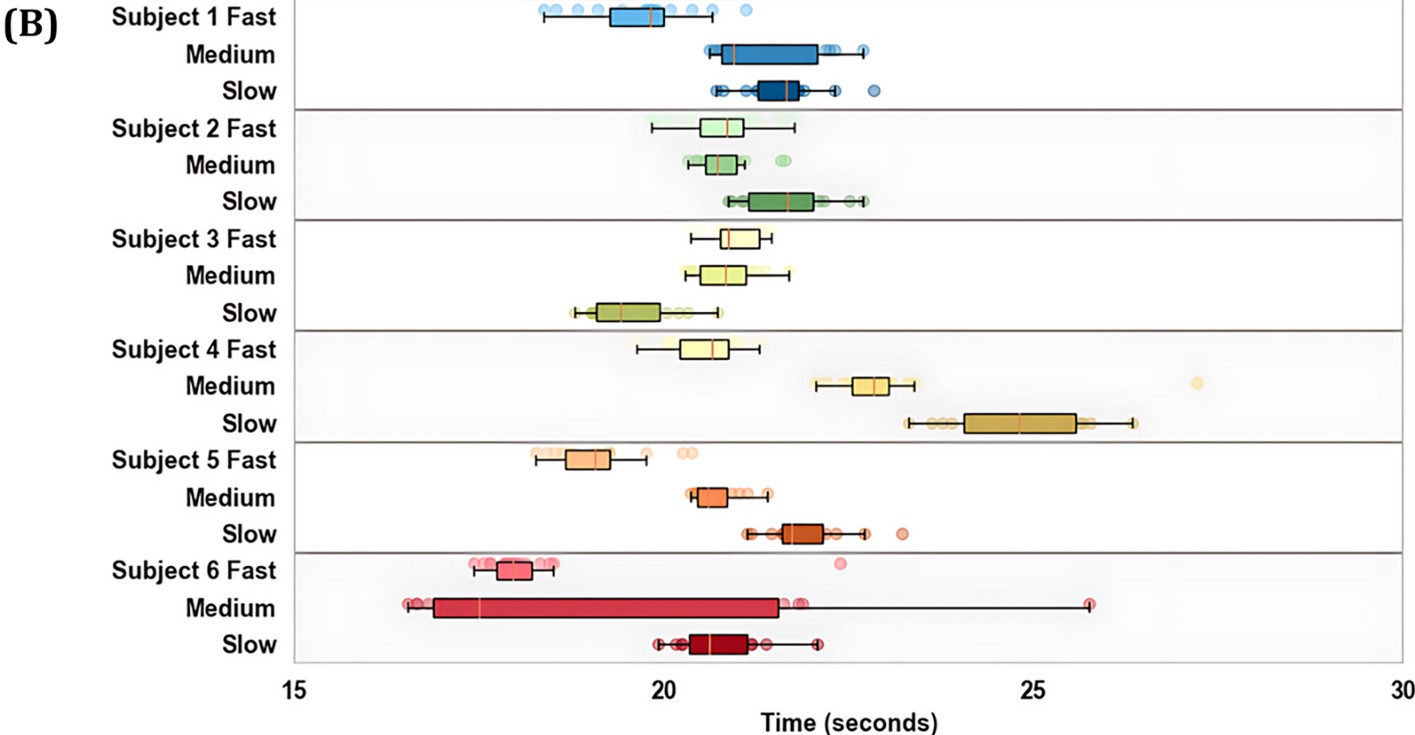

**Figure 4 Simulation times separated by participant and speed for optimal control simulations (A) and IMU-based inverse kinematics (B).**

**Table 2 Raw (A) and normalised (B) RMSE (±STD) for each joint and speed between motion capture and optimal control simulations (OC) or IMU-based inverse kinematics (IK).** Maximum values for each condition are highlighted in bold.

| | Slow | | Medium | | Fast | | Average | |
|---|---|---|---|---|---|---|---|---|
| | OC | IK | OC | IK | OC | IK | OC | IK |
| **(A)** | | | | | | | | |
| Pelvis tilt | 4.84 ± 0.74 | 6.99 ± 0.53 | 4.89 ± 0.88 | 6.83 ± 0.57 | 5.07 ± 1.14 | 6.81 ± 1.01 | 4.93 ± 0.92 | 6.88 ± 0.70 |
| Pelvis list | 5.62 ± 0.67 | 5.83 ± 0.41 | 5.62 ± 0.72 | 5.72 ± 0.46 | 5.82 ± 0.76 | 5.71 ± 0.65 | 5.69 ± 0.72 | 5.75 ± 0.51 |
| Pelvis rotation | 3.87 ± 0.79 | 4.21 ± 0.60 | 4.27 ± 0.97 | 4.34 ± 0.63 | 4.28 ± 0.87 | 4.39 ± 0.68 | 4.14 ± 0.88 | 4.31 ± 0.64 |
| Lumbar extension | **12.41 ± 1.48** | 7.21 ± 0.70 | **12.43 ± 1.69** | 7.37 ± 0.74 | **12.43 ± 1.81** | 7.63 ± 1.00 | **12.42 ± 1.66** | 7.40 ± 0.81 |
| Lumbar bending | 5.11 ± 0.72 | 8.37 ± 0.62 | 5.25 ± 1.00 | 8.02 ± 0.61 | 5.42 ± 1.20 | 7.93 ± 1.01 | 5.26 ± 0.97 | 8.11 ± 0.75 |
| Lumbar rotation | 5.91 ± 1.14 | 8.71 ± 0.73 | 5.99 ± 1.39 | 8.30 ± 0.73 | 6.14 ± 1.41 | 8.18 ± 0.99 | 6.01 ± 1.31 | 8.40 ± 0.82 |
| Hip flexion | 9.15 ± 0.75 | 15.77 ± 0.66 | 9.34 ± 1.10 | 14.60 ± 0.70 | 9.68 ± 1.54 | 13.74 ± 1.35 | 9.39 ± 1.13 | 14.70 ± 0.90 |
| Hip adduction | 5.85 ± 0.69 | 6.56 ± 0.45 | 5.97 ± 0.78 | 6.41 ± 0.47 | 6.12 ± 0.87 | 6.37 ± 0.59 | 5.98 ± 0.78 | 6.45 ± 0.50 |
| Hip rotation | 6.89 ± 0.83 | 7.89 ± 0.59 | 7.53 ± 1.47 | 7.91 ± 0.65 | 7.53 ± 1.42 | 7.92 ± 0.87 | 7.32 ± 1.24 | 7.91 ± 0.70 |
| Knee angle | 8.08 ± 0.77 | **25.87 ± 0.85** | 8.19 ± 1.05 | **23.33 ± 0.89** | 8.57 ± 1.37 | **21.56 ± 0.77** | 8.28 ± 1.06 | **23.59 ± 0.84** |
| Ankle angle | 11.90 ± 0.59 | 14.62 ± 0.60 | 12.10 ± 0.78 | 14.82 ± 0.61 | 12.39 ± 0.95 | 15.05 ± 0.89 | 12.13 ± 0.77 | 14.83 ± 0.70 |
| **Mean** | 7.24 ± 0.83 | 10.18 ± 0.61 | 7.42 ± 1.08 | 9.79 ± 0.64 | 7.59 ± 1.21 | 9.57 ± 0.89 | 7.42 ± 1.04 | 9.85 ± 0.71 |
| **(B)** | | | | | | | | |
| Pelvis tilt | 0.41 ± 0.06 | **0.6 ± 0.05** | 0.4 ± 0.07 | **0.57 ± 0.05** | 0.4 ± 0.09 | **0.55 ± 0.08** | 0.4 ± 0.07 | **0.57 ± 0.06** |
| Pelvis list | 0.33 ± 0.04 | 0.35 ± 0.02 | 0.32 ± 0.04 | 0.33 ± 0.03 | 0.33 ± 0.04 | 0.33 ± 0.04 | 0.33 ± 0.04 | 0.34 ± 0.03 |
| Pelvis rotation | 0.29 ± 0.06 | 0.31 ± 0.05 | 0.3 ± 0.07 | 0.3 ± 0.04 | 0.29 ± 0.06 | 0.29 ± 0.05 | 0.29 ± 0.06 | 0.3 ± 0.05 |
| Lumbar extension | **0.74 ± 0.09** | 0.41 ± 0.04 | **0.7 ± 0.09** | 0.4 ± 0.04 | **0.67 ± 0.09** | 0.39 ± 0.05 | **0.7 ± 0.09** | 0.4 ± 0.04 |
| Lumbar bending | 0.21 ± 0.03 | 0.35 ± 0.03 | 0.21 ± 0.04 | 0.33 ± 0.03 | 0.21 ± 0.05 | 0.32 ± 0.04 | 0.21 ± 0.04 | 0.33 ± 0.03 |
| Lumbar rotation | 0.17 ± 0.03 | 0.24 ± 0.02 | 0.16 ± 0.04 | 0.23 ± 0.02 | 0.16 ± 0.04 | 0.22 ± 0.03 | 0.16 ± 0.04 | 0.23 ± 0.02 |
| Hip flexion | 0.17 ± 0.01 | 0.29 ± 0.01 | 0.16 ± 0.02 | 0.26 ± 0.01 | 0.16 ± 0.03 | 0.23 ± 0.02 | 0.16 ± 0.02 | 0.26 ± 0.01 |
| Hip adduction | 0.23 ± 0.03 | 0.26 ± 0.02 | 0.23 ± 0.03 | 0.25 ± 0.02 | 0.23 ± 0.03 | 0.24 ± 0.02 | 0.23 ± 0.03 | 0.25 ± 0.02 |
| Hip rotation | 0.46 ± 0.06 | 0.53 ± 0.04 | 0.48 ± 0.09 | 0.5 ± 0.04 | 0.46 ± 0.09 | 0.48 ± 0.05 | 0.47 ± 0.08 | 0.5 ± 0.04 |
| Knee angle | 0.09 ± 0.01 | 0.28 ± 0.01 | 0.09 ± 0.01 | 0.24 ± 0.01 | 0.09 ± 0.01 | 0.22 ± 0.01 | 0.09 ± 0.01 | 0.25 ± 0.01 |
| Ankle angle | 0.24 ± 0.01 | 0.3 ± 0.01 | 0.24 ± 0.02 | 0.3 ± 0.01 | 0.25 ± 0.02 | 0.3 ± 0.02 | 0.24 ± 0.02 | 0.3 ± 0.01 |
| **Mean** | 0.3 ± 0.04 | 0.36 ± 0.03 | 0.3 ± 0.05 | 0.34 ± 0.03 | 0.3 ± 0.05 | 0.33 ± 0.04 | 0.3 ± 0.05 | 0.34 ± 0.03 |

pelvic tilt could be attributed to differences in static postures when placing the IMUs on participants. Given the lack of consistent offset observed across participants, method uniformity was prioritized over individual correction factors.

A similar offset pattern emerged for the ankle flexion angle during the stance phase. The difference here may reflect the simulation's attempts to employ appropriate contact mechanics to appropriately balance forces and facilitate forward running propulsion, rather than adhering to tracking foot IMU signals. In contrast, the inverse simulations did not consider contact tracking with the ground; hence ankle angles were dictated only by kinematic constraints and the tracking of sensor orientations, allowing the foot to penetrate the ground. This observation highlights a trade-off between realistic contact

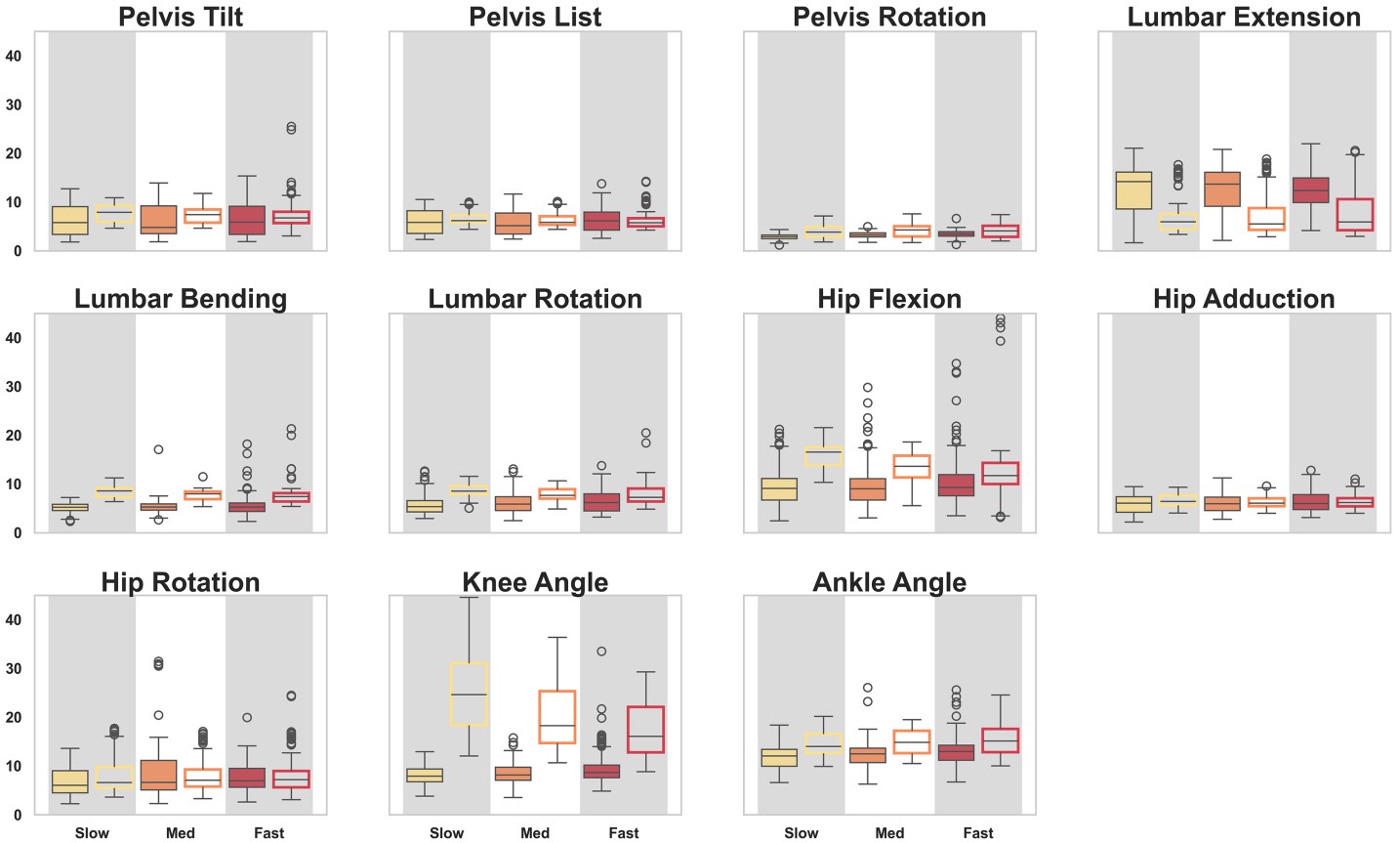

**Figure 5** RMSE between IMU-based optimal control simulations (filled) and marker-based inverse kinematics (outline) methods with optical motion capture over each joint angle and speed.

modelling and the representation of true joint angles without considering external loads. Although better results may have been obtained in the optimal control simulations by tracking measured ground reaction forces rather than generic forces scaled with body weight, the requirement for force plates in conjunction with IMUs is less translatable to real-world data collection.

Past research comparing IMU kinematics with motion capture methods in running gait is limited when considering an inverse kinematics or optimal control approach. For inverse kinematics, the RMSE in two studies ranged from 5–8° (*Lin et al., 2023*) and 18–28°, reduced to 5–8° with an offset correction (*Nüesch et al., 2017*). In this study, no offset correction was applied, and comparable error magnitudes (4–26°) were observed in IMU-based inverse kinematics. In the case of optimal control, errors ranging from 5–9° have been reported (*Dorschky et al., 2019b*), typically increasing when fewer sensors are used (*Dorschky et al., 2024*) and consistent with the errors observed in this study (4–12°). The aforementioned study revealed increasing accuracy in more distal joints, unlike the findings of this study. The improved ankle flexion accuracy could be a consequence of the use of variable temporal weighting factors. Periods with large artefacts in the signal, such as during foot strike, would ideally not be tracked as closely in the optimal control

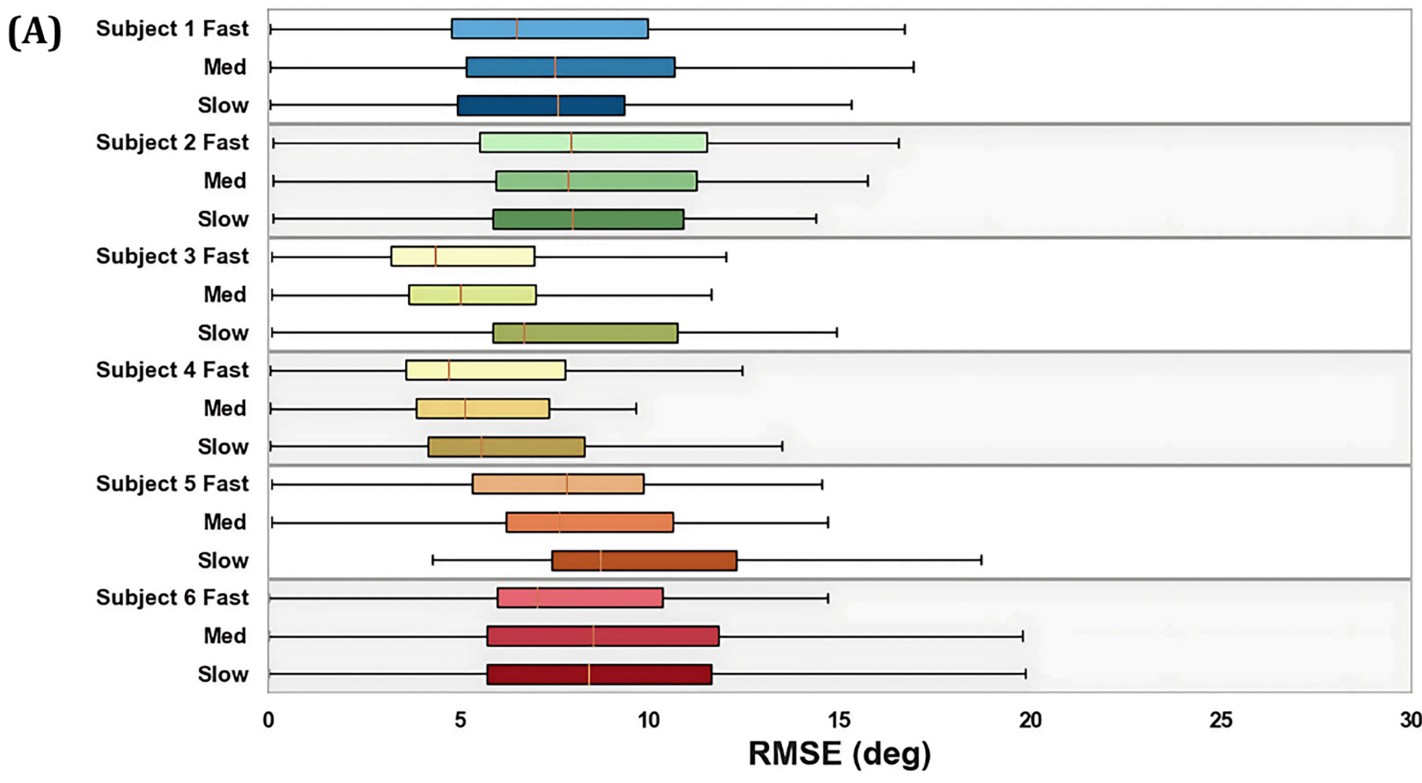

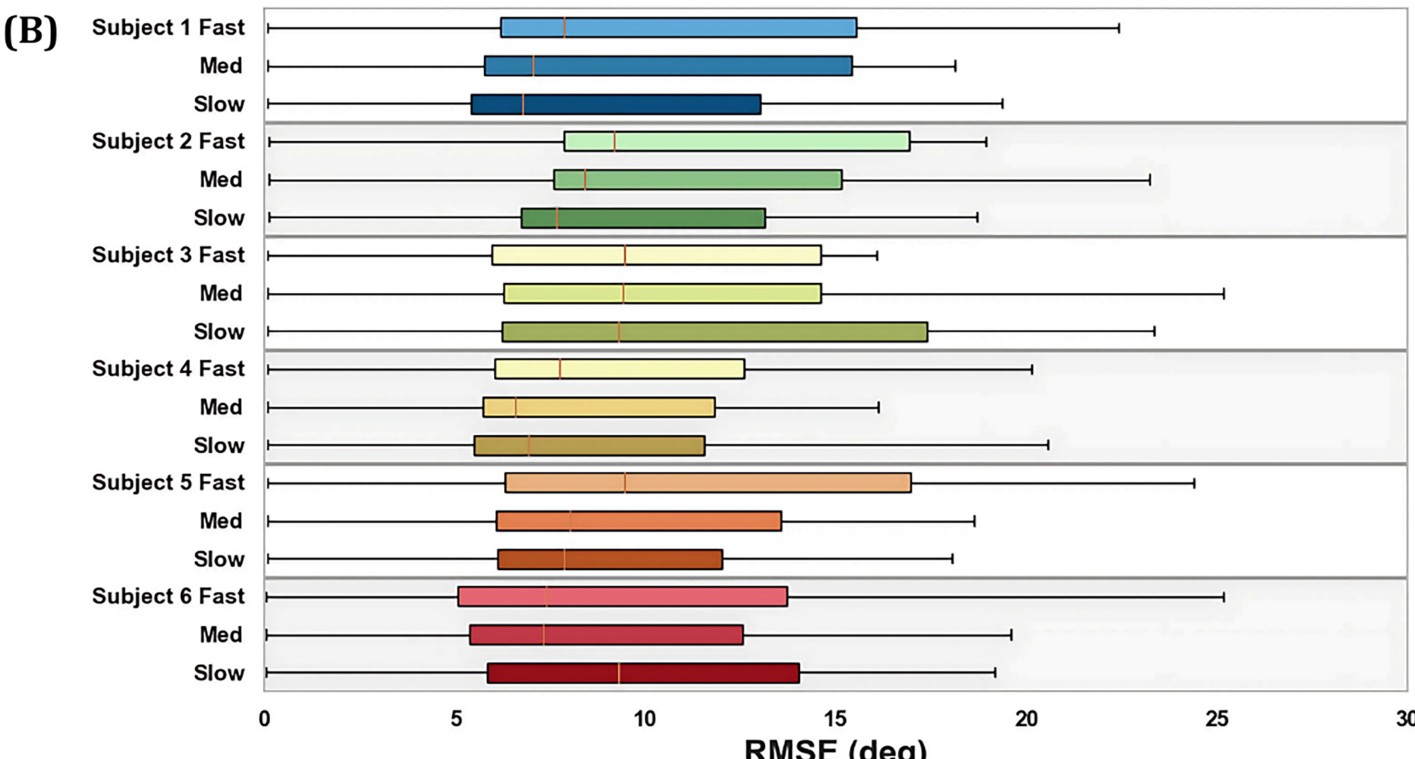

**Figure 6** Box plots of the RMSE with marker-based inverse kinematics for both optimcal control simulations (A) and IMU-based inverse kinematics (B), separated by participant and speed over all joint angles.

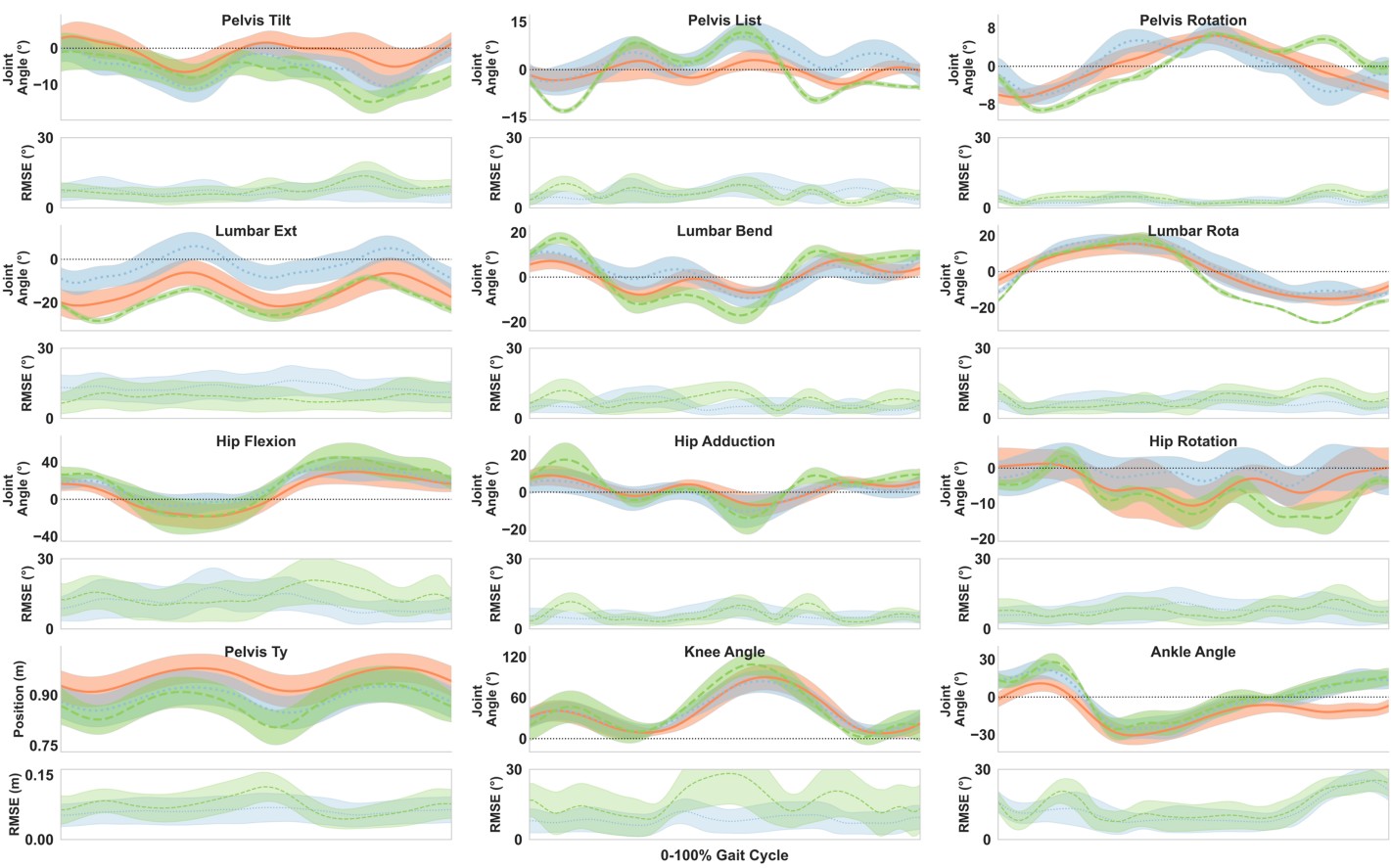

**Figure 7 Joint kinematics between marker-based motion capture (orange), optimal control simulations (blue) and IMU-based inverse kinematics (green).** RMSE is plotted below each joint, with shading indicating the standard deviation in absolute error.

simulations, however, a temporal change in weighting factors was not a feature currently available in the optimal control software.

Machine learning methods are increasingly popular alternatives to both inverse kinematics and optimal control simulations for deriving biomechanical outcomes from IMU sensor data without relying on explicitly defined biomechanical models (*Xiang et al., 2022*). These methods have demonstrated accurate kinematic predictions across various movement types, often requiring fewer sensors and showing robustness to noise (*Dorschky et al., 2020*; *Hernandez et al., 2021*; *Rapp et al., 2021*; *Xiang et al., 2022*; *García-de-Villa et al., 2023*; *Gholami, Napier & Menon, 2020*). However, machine learning approaches also face challenges, including the need for large, high-quality datasets for training, the potential for over-fitting, and models trained only specific to a particular motion, population, or individual (*Gurchiek, Cheney & McGinnis, 2019*). Unlike a musculoskeletal modelling approach, these methods typically only evaluate a limited range of outcomes in one domain, such as kinematics, ground reaction forces, or joint forces (*Xiang et al., 2022*). While machine learning can eliminate the need for detailed input parameters, it typically lacks the direct insight that biomechanical models can provide, limiting its interpretability

for understanding underlying movement strategies (*Gurchiek, Cheney & McGinnis, 2019*; *Rodu et al., 2024*).

Marker-based motion capture is considered the reference standard; however, it is pertinent to acknowledge the inherent errors associated with this method, which could impact the accuracy of the study findings when compared to true joint angles. Motion capture systems can have errors of 1–5 mm, *Chiari et al. (2005)* compounded by soft tissue artefacts of up to 10 mm (*Leardini et al., 2005*). The marker registration error further contributes to this uncertainty; a 2 cm difference resulted in a variation in peak ankle, knee, and hip sagittal angles of up to 15° (*Uchida & Seth, 2022*). The finding that the resulting joint angles were within the bounds of other experimental data provides further confidence in the study findings in light of this uncertainty. Ultimately, the modelling relevance of this study lies in the ability to measure gait changes within and between individuals, which were observed across participants at different running speeds.

A key strength of this modelling approach lies in the incorporation of a three-dimensional model in both methods. Modelling outside the sagittal plane could capture more intricate aspects of human locomotion, such as hip adduction and foot pronation, which can be particularly pertinent in running gait analysis. The ability of the model to capture these out-of-plane movements in optimal control simulations was demonstrated, with no region having an RMSE greater than 10°. Moreover, the multi-segment foot model used in the optimal control simulations allowed the MTP joint's role during the push-off phase of running gait to be accounted for, enhancing the realism of the simulations and ground contact modelling (*Falisse, Afschrift & Groote, 2021*).

A limitation lies in the use of a torque-driven model in the optimal control simulations. These models lump synergistic muscle forces together and assume that the maximal torque exerted at a joint is a function of the kinematics of that joint alone. Torque-driven simulations typically have fewer unknown parameters compared to individual muscle-model simulations. It is feasible that this approach could be extended to incorporate muscle-driven simulations, where the optimization process includes muscle dynamics and activation patterns to more accurately represent the underlying physiology. This was avoided, however, due to the significant increase in computational cost and simulation complexity involved in simulating muscle dynamics, and remains an area for future improvement.

A further limitation lies in the potential generalizability of the approach. The small sample size in this study restricts the applicability of the results to broader populations, particularly those with varying anthropometric characteristics or gait patterns. Moreover, while the approach has been validated for running gait, its application to other dynamic activities or non-linear movement patterns remains unexplored. Additionally, the controlled laboratory environment may not fully capture the variability present in real-world conditions. Although the method appears translatable to field settings, it has not yet been tested in such environments.

The sensor setup employed in this study consisted of eight sensors. The IMU tracking strategy using optimal control could be adapted to situations with fewer IMUs, which offers advantages in cost and convenience (*Dorschky et al., 2024*). In such cases,

emphasizing the minimization of effort and tracking generic running kinematics is likely needed to compensate for body segments lacking IMUs. Fewer sensors could thus make the motion more generalised, limiting the ability to model unique gait patterns. Given that the goal of this study was to achieve optimal agreement with subject-specific motion capture kinematics, all available sensors were utilized in the simulations. The IMU-based inverse kinematics approach, which is solely dependent on the sensor orientations to estimate joint angles, is unlikely to be adaptable to involving fewer sensors for gait analysis.

The greater resolution of optimal control simulations comes at the expense of computational time and user input. On average, the simulations took nearly 150 times the duration of IMU-based inverse kinematics (Fig. 4). Not accounted for is the additional time spent iteratively testing the simulation parameters to ensure convergence on a feasible solution. Unlike inverse methods, optimal control presents limitless inputs, including the initial guess, weighting factors, and constraint combinations. While the ability of these simulations to more closely replicate gold-standard kinematics underscores the potential of optimal control in biomechanical modelling, it is important to acknowledge that this setup remains somewhat subjective, without a universal method of application. The consistent application of weighting factors and constraints across participants and speeds demonstrates the robustness of the approach used in this study. However, critical questions regarding the integrity of the control problem and the influence of input parameters on outcomes remain unanswered. As such, the intricate nature of optimal control may not align with the practicality and ease of use required for everyday individuals to deploy IMU sensors effectively in field settings, posing barriers to widespread adoption and implementation of this approach without further development.

On the other hand, inverse kinematics, both IMU-based and marker-based, is computationally simpler to implement and solve than optimal control methods, with significantly shorter computational times and fewer input parameters. These methods assume that joint motion can be accurately determined solely based on endpoint positions, neglecting the underlying dynamics of the system. This simplification may lead to less accurate representations of joint motion and its associated drivers, particularly in dynamic tasks or when dealing with complex movement patterns. The choice between these approaches depends on the specific research questions, available time and resources, and desired level of detail and accuracy in the modelling process.

The findings of this study have important implications for the field of biomechanics and the assessment of human movement. Optimal control simulations not only outperformed IMU-based inverse kinematics in accuracy in this study, but also offer distinct advantages in biomechanical modeling over both inverse and machine learning approaches. These simulations enable the simultaneous estimation of outcomes such as ground reaction forces, muscle activations, joint loads, and kinematics through a single trajectory optimization that adheres to physiological and physical principles. Beyond data tracking, optimal control methods allow a wide range of objectives to be considered, including optimality principles underlying gait (Miller et al., 2012; De Groote & Falisse, 2021). In addition to the tracking of IMU signals, the cost function in this study included goals related to minimizing effort, enforcing joint angle symmetry, and, to a lesser extent,

tracking generic kinematics and ground reaction force trajectories. Aligning the modelling approach with the fundamental objectives of locomotion reduces the limitations associated with traditional IMU-based methods. Reducing the dependence on potentially error-prone experimental IMU data mitigates inaccuracies in the step-wise computation of outcome variables. In addition, by optimizing control inputs (muscles or torque actuators), these simulations provide insights into how activation patterns and strategies achieve specific movements, making them particularly effective at capturing the complex interactions of forces and dynamics in human movement. Future research should continue to explore the potential of optimal control methods with inertial sensors, encompassing diverse locomotion forms, populations, and clinical and sporting applications. Efforts to refine the objective function and associated weighting factors used in optimal control simulations could further enhance model accuracy and validity.

## CONCLUSION

This study demonstrated the ability of two modelling methods using IMUs to reproduce joint kinematics, with closer agreement to optical marker-based motion capture observed with optimal control modelling. The use of a three-dimensional, torque-driven biomechanical model and a custom model-scaling method allowed for a precise and comprehensive representation of gait in a single dynamic optimal control simulation. An optimal control paradigm enhances the robustness of IMU data to noise and errors, presenting a promising solution to mitigate limitations seen in traditional inverse-based approaches that rely heavily on and are more susceptible to errors in experimental data. Inverse kinematics provides a more straightforward and computationally efficient solution, making it more suitable for real-time applications and situations where computational resources are limited. Optimal control simulations also integrate dynamic constraints, providing a more comprehensive understanding of the underlying forces and activations driving the observed movements. This approach, when integrated with IMUs, opens new possibilities for advanced biomechanical modelling methods, such as modelling metabolic cost or muscle activation dynamics.

### Funding
The authors received no funding for this work.

### Competing Interests
The authors declare that they have no competing interests.

### Author Contributions
- Grace McConnochie conceived and designed the experiments, performed the experiments, analyzed the data, prepared figures and/or tables, authored or reviewed drafts of the article, and approved the final draft.
- Aaron S. Fox conceived and designed the experiments, authored or reviewed drafts of the article, and approved the final draft.

- Clint Bellenger conceived and designed the experiments, authored or reviewed drafts of the article, and approved the final draft.
- Dominic Thewlis conceived and designed the experiments, performed the experiments, authored or reviewed drafts of the article, and approved the final draft.

## Human Ethics

The following information was supplied relating to ethical approvals (*i.e.*, approving body and any reference numbers):

The University of Adelaide Low Risk Human Research Ethics Review Group (Faculty of Health and Medical Sciences) granted Ethical approval to carry out the study within its facilities. ETHICS APPROVAL No: H-2022-120.

## Data Availability

The data is available at GitHub and Zenodo:

- https://github.com/GraceEMc/IMURunKinematics

- GraceEMc. (2025). GraceEMc/IMURunKinematics: PeerJ (Version publish2).
Zenodo. https://doi.org/10.5281/zenodo.14796986.

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
