# Peer review of "Optimal control simulations tracking wearable sensor signals provide comparable running gait kinematics to marker-based motion capture"

_PeerJ, doi:10.7717/peerj.19035_

## Round 0.1 · original submission · Minor Revisions

Fine work!

Please address the reviewers' comments.

Reviewer 1 ·

Basic reporting

The document is mainly clear, but some improvements should be made. There are some typographical mistakes that need to be eliminated, such as spaces before of after parenthesis (e.g. line 61, line 97), lack of dots (after i.e. in line 355) or singular words that should be plural (e.g. line 62). Please, check the last citation in line 97. Please use the word degrees or its symbol, but not change it along the document. Some sentences are not clear, see lines 119 after the dot (it seems to be repeating the previous sentence) and lines 338-339, but reread and clarify the document.

With regard to the organization of tables and figures, they should be right after the first paragraph in which they are cited, because otherwise reading the document is confusing. Please, reorganize them.

To support the introduction of your work (lines 52-54, and 64-69, among others), there are more recent works, such as the following:
García-de-Villa, S., Casillas-Pérez, D., Jiménez-Martín, A., & García-Domínguez, J. J. (2023). Inertial sensors for human motion analysis: A comprehensive review. IEEE Transactions on Instrumentation and Measurement, 72, 1-39. https://ieeexplore.ieee.org/abstract/document/10124779
And the citations included in that work. This review also supports the low number of participants in your study, which is in line with the average number of participants for these kind of IMU-based kinematic analysis proposals.
Moreover, that work summarizes other works which estimate human kinematics based on machine learning (shallow and deep) and given their relevance and the growing tendency of their use, these methods should be more deeply introduced and discussed (not just cited) in your work as an alternative to the proposed approaches.

Experimental design

This work evaluates two approaches to obtain three dimensional lower-limb kinematics in running by using inertial measurement units (IMUs). The validation of the proposed methods is carried out by comparison with an optical marker-based motion capture system that uses the inverse kinematics solver in Opensim to estimate the joint angles. The first proposed approach is based on inverse kinematics and the second proposal on optimal control simulations. Both proposals use OpenSim. The hypothesis and experiments are well designed and relevant. The manuscript is well written and structured.

Please find some questions about your research in the following:

Line 184: which approach do you use to fill gaps due to occlusions in the data? Which is the criteria to remove data points?
Line 199-190: are magnetometer data not filtered?
Line 267: when you say the scaled model is consistent with the one used in the inverse method, you mean it is the same model? Are there any differences?

Line 310: parameter distance D, which distance is it referring to?

Line 317: as far as I understand about this line, the optimal control approach uses as inputs some of the results from the IK analysis because it improves its results. However, these results are not included in this work. They should be included, at least in an appendix for the reader to understand this discussion in the approach. Also, why are these angles the only ones to be included in this optimization method? For example, the lumbar flexion DOF provides high errors that might be improved with this angle from the IMU inverse kinematics solved for in OpenSense.
Related with the previous query about the flowchart, this relationship of methods should be clarified in the aforementioned flowchart.

Validity of the findings

The research is solid and provides interesting results with relevant implications for biomechanics researching. The result section would be clearer if, the figures and tables are reorganized.

In the discussion, it would be easier to follow if it included more references to the previous tables and figures in the result section.

It would be interesting to compare the results and ventages and disadvantages (accuracy, number of sensors, generalization capacity, etc.) of the proposals with machine learning methods should be included, given the relevance of these methods.

Additional comments

Since the proposal involve some steps for preprocessing and processing, the methods sections should be enriched with a flowchart that completely describes the proposal. In this method, the OpenSim-based processing could be highlighted.

Line 239: please check the expression for the mode-predicted position.
Line 241: please check that MTP acronym is defined.

Line 256: please specify which is the information to see in the supplementary file. Same comment for line 337, which also refers to the supplementary file. Also avoid writing the reference to the supplementary file out of a sentence and include this reference as a part of the text.

Line 271: please check the units of the magnitudes given.
Line 302: N should be N1. The same happens in 316, where it should be N2.
Lines 306-307 and lines 313-315 are not a sentence, please rewrite so that it is.

Line 392: RMSE acronym already used, please check it is defined in its first use.
Line 453: please check the citation style of this reference.

·

Basic reporting

The writing is clear, precise, and professional, with impeccable English. The introduction and background provide appropriate context, and the literature review is relevant and well-referenced. The work from McConnochie et al. manage to communicate their in-depth analysis of the biomechanical modeling of running in an accessible way.
No further comment

Experimental design

The experimental design is original and falls within the scope of the journal. The research question is well-defined, and meaningfully addresses an identified knowledge gap: validity and limitations of data obtained via wearable IMUs for the kinematic running analysis.

Validity of the findings

Regarding the validity of the findings, the data are robust, statistically sound, and controlled. To some extent, the impact of the study may be minimized by the small sample size, and perhaps with only six participants, it would have been more desirable to select them from a similar performance level so that the same running speeds were included (making a normative database on running movement available in the medium/long term).
In any case, the novelty of the model used and the way the authors describe their methodology clearly encourages replication of their work, and the benefits to biomechanics and the science of running are clearly stated.

Additional comments

Minor comments:
Please remove “49” from the last paragraph in the abstract (only in the first page)
Line 417: Please correct the word “contract”
Did the researchers control the use of supershoes (i.e., running shoes with carbon fiber plates) during the evaluation?
I agree with the authors that adding muscle activation data through electromyography could open up a very interesting future direction to continue advancing with their model. Additionally, including multiple IMUs on the upper limb body segments would complete the analysis of the runner's kinematics as a whole and help to better understand the dynamics in all planes.

Reviewer 3 ·

Basic reporting

Line 54: Suggest rephrasing “…and optionally magnetometer.”

Need to proof read manuscript and fix grammatical and other errors. Examples:
Line 68: Comma out of place, and awkward wording re: “with these gait observations”

Line 79: Incomplete sentence, out of place hyphen in middle of sentence, and spelling

Line 96: Includes citation title in one reference

Experimental design

What was the rationale for the inclusion criteria of running a recent race? Especially in the context of the presented analyses/aims for the manuscript, it’s not clear why this would be important.

It is standard practice to describe the Methods in sequential order unless there is some logical reason to not do this. In this case, it seems to make more sense to put them in sequential order, re: describing the participant instrumentation prior to the actual procedures they performed.

Lines 193-200: I may be misinterpreting this, so please correct me if way off: This section makes it seem as if your methods for estimating joint kinematics from the sensors (both methods) are dependent on aligning the sensor axes to each respective segment, using the static trial optical motion capture data. This is a shortcoming I’ve seen in several other approaches, where the whole rationale behind IMU-based kinematics is to be “ecologically valid” and allow for collection outside of a laboratory without optical motion capture. But if the methods rely on optical motion capture to orient the sensors, then how is this really true?

Lines 352-357: So in essence, you threw out any trials for the optimal control method that were really bad? Did this only involve the 3 outliers mentioned below for Subject 3? If so, this may not be a big issue, but the point could be made that you would have no idea which trials were outliers in a real-world scenario, i.e., when the optical motion capture data is not available. So how would you be able to use this method with confidence?

Could you comment on why you chose to only assess sagittal plane variables for the knee/ankle? At the very least, frontal plane variables for both joints are highly relevant during running, especially in terms of loading/stress to musculoskeletal tissues and the occurrence of running-related musculoskeletal injury.

Validity of the findings

Line 394: How do you reconcile this fact with previous literature that has consistently shown that the best estimations for kinematics are usually for sagittal joint angles? This is largely true of any motion capture system/method, including optical moCap.

Lines 425-427: I’d first question whether this is really an exhaustive list of studies that have evaluated IMU-based kinematics during running. Below is at least one additional study, and I believe there are more.

Second, some discussion of why the inverse dynamics based IMU approach in the current study showed such disparate results compared to previous work seems to be warranted. The authors sort of skim past this, where it naturally raises the question of whether the optimal control approach was really that superior, or if there was some issue with the comparative approach in the current study. Especially considering the previous comment on the sagittal joint angles being the worst, which is not typical. A mean error of 23 degrees for knee flexion angles in the current study, compared to 5-8 in previous studies and even less in the additional one mentioned below, seems like a big discrepancy.

• Hernandez, V., Dadkhah, D., Babakeshizadeh, V., & Kulić, D. (2021). Lower body kinematics estimation from wearable sensors for walking and running: A deep learning approach. Gait & posture, 83, 185-193.

There also seems to be a need for some discussion of whether the optimal control approach provides adequate accuracy, outside of comparing it to the inverse dynamics approach. Especially considering the high relative errors (most 20-70% of the joint ROM) and the lengthy computation time, does it even seem like this method is useful, even if it is more accurate than the comparison?

For these last two comments, I think that much of the Discussion can be shortened up or removed to make more room for the added language. See below for recommendation on a paragraph to remove. I would suggest going through and identifying similar language that isn’t directly focused on explaining the results or their impact, and shorten it up further.

Additional comments

Keep consistent language, re: OpenSim/Moco vs. IMU-based inverse kinematics/optimal control simulations. The latter is used more frequently, so recommend changing all language to this.

Figure 4 replicates Table 2 (i.e., same data presented two different ways). Recommend simply removing Figure 4. I’m also not really sure what Figure 5 adds to the interpretation of results, especially because it’s hard to directly compare the two methods.

Table 2- Need to indicate what the values to the right of ‘+/-‘ are

Lines 473-483: Recommend removing this paragraph. This seems to fit better in the Introduction, but you already have some language discussing the theoretical benefits of optimal control simulations there.

---

## Round 0.2 · accepted · Accept

Both reviewers were sufficiently satisfied with the revisions.

I concur. I'm intrigued by the topic of the paper and believe it will be a nice addition to scientific literature.

·

Basic reporting

I would like to express my sincere appreciation for the honesty and clarity with which the authors addressed the concerns raised in my initial review of this manuscript. The responses were comprehensive and well-explained, providing valuable insights into the revisions made.

In my opinion, the manuscript is now in a much stronger position, and I believe it is ready to move forward in the processing stage. I commend the authors´ efforts and look forward to seeing the continued development of their work.

Experimental design

No further comments

Validity of the findings

Thanks for the clarifications and for acknowledging the limitations of the study in terms of the generalization of its results in the discussion.

Additional comments

No further comments

Reviewer 3 ·

Basic reporting

--

Experimental design

--

Validity of the findings

--